# Improving Robustness Without Sacrificing Accuracy with Patch Gaussian Augmentation

## Abstract

Deploying machine learning systems in the real world requires both high accuracy on clean data and robustness to naturally occurring corruptions. While architectural advances have led to improved accuracy, building robust models remains challenging, involving major changes in training procedure and datasets. Prior work has argued that there is an inherent trade-off between robustness and accuracy, as exemplified by standard data augmentation techniques such as `Cutout`, which improves clean accuracy but not robustness, and additive `Gaussian` noise, which improves robustness but hurts accuracy. We introduce `Patch Gaussian`, a simple augmentation scheme that adds noise to randomly selected patches in an input image. Models trained with `Patch Gaussian` achieve state of the art on the CIFAR-10 and ImageNet Common Corruptions benchmarks while also maintaining accuracy on clean data. We find that this augmentation leads to reduced sensitivity to high frequency noise (similar to `Gaussian`) while retaining the ability to take advantage of relevant high frequency information in the image (similar to `Cutout`). We show it can be used in conjunction with other regularization methods and data augmentation policies such as AutoAugment. Finally, we find that the idea of restricting perturbations to patches can also be useful in the context of adversarial learning, yielding models without the loss in accuracy that is found with unconstrained adversarial training.

## 1 Introduction

Modern deep neural networks can achieve impressive performance at classifying images in curated datasets (Karpathy, 2011; Krizhevsky et al., 2012; Tan & Le, 2019). Yet, they lack robustness to various forms of distribution shift that typically occur in real-world settings. For example, neural networks are sensitive to small translations and changes in scale (Azulay & Weiss, 2018), blurring and additive noise (Dodge & Karam, 2017), small objects placed in images (Rosenfeld et al., 2018), and even different images from a distribution similar to the training set (Recht et al., 2019; 2018). For models to be useful in the real world, they need to be both accurate on a high-quality held-out set of images, which we refer to as "clean accuracy," and robust on corrupted images, which we refer to as "robustness." Most of the literature in machine learning has focused on architectural changes (Simonyan & Zisserman, 2015; Szegedy et al., 2015; He et al., 2016; Zoph & Le, 2017; Szegedy et al., 2017; Han et al., 2017; Zoph et al., 2017; Hu et al., 2017; Liu et al., 2018) to improve clean accuracy but interest has recently shifted toward robustness as well.

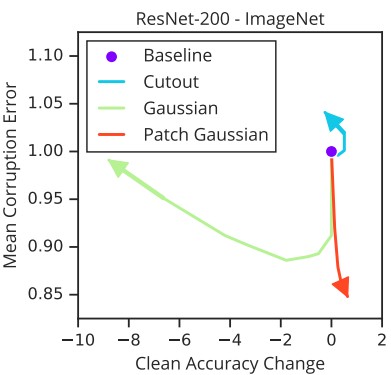

Figure 1: `Patch Gaussian` augmentation overcomes the accuracy/robustness trade-off observed in other augmentation strategies. Larger $\sigma$ of `Patch Gaussian` (→) improves mean corruption error (mCE) and maintains clean accuracy, whereas larger $\sigma$ of `Gaussian` (→) and patch size of `Cutout` (→) hurt accuracy or robustness. More robust and accurate models are down and to the right.

Research in neural network robustness has tried to quantify the problem by establishing benchmarks that directly measure it (Hendrycks & Dietterich, 2018; Gu et al., 2019) and comparing the performance of humans and neural networks (Geirhos et al., 2018b; Elsayed et al., 2018). Others have tried

to understand robustness by highlighting systemic failure modes of current methods. For instance, networks exhibit excessive invariance to visual features (Jacobsen et al., 2018), texture bias (Geirhos et al., 2018a), sensitivity to worst-case (adversarial) perturbations (Goodfellow et al., 2014), and a propensity to rely on non-robust, but highly predictive features for classification (Doersch et al., 2015; Ilyas et al., 2019). Of particular relevance, Ford et al. (2019) has established connections between popular notions of adversarial robustness and some measures of distribution shift considered here.

Another line of work has attempted to increase model robustness performance, either by projecting out superficial statistics (Wang et al., 2019), via architectural improvements (Cubuk et al., 2017), pre-training schemes (Hendrycks et al., 2019), or with the use of data augmentations. Data augmentation increases the size and diversity of the training set, and provides a simple way to learn invariances that are challenging to encode architecturally (Cubuk et al., 2017). Recent work in this area includes learning better transformations (DeVries & Taylor, 2017; Zhang et al., 2017; Zhong et al., 2017), inferring combinations of them (Cubuk et al., 2018), unsupervised methods (Xie et al., 2019), theory of data augmentation (Dao et al., 2018), and applications for one-shot learning (Asano et al., 2019).

Despite these advances, individual data augmentation methods that improve robustness do so at the expense of reduced clean accuracy. Further, achieving robustness on par with the human visual system is thought to require major changes in training procedures and datasets: the current state of the art in robustness benchmarks involves creating a custom dataset with styled-transferred images before training (Geirhos et al., 2018a), and still incurs a significant drop in clean accuracy. The ubiquity of reported robustness/accuracy trade-offs in the literature have even led to the hypothesis that these trade-offs may be inevitable (Tsipras et al., 2018). Because of this, many recent works focus on improving either one or the other (Madry et al., 2017; Geirhos et al., 2018a). In this work we propose a simple data augmentation method that overcomes this trade-off, achieving improved robustness while maintaining clean accuracy. Our contributions are as follows:

- We characterize a trade-off between robustness and accuracy in standard data augmentations `Cutout` and `Gaussian` (Section 2.1).
- We describe a simple data augmentation method (which we term `Patch Gaussian`) that allows us to interpolate between the two augmentations above (Section 3.1). Despite its simplicity, `Patch Gaussian` achieves a new state of the art in the Common Corruptions robustness benchmark (Hendrycks & Dietterich, 2018), while maintaining clean accuracy, indicating current methods have not reached this fundamental trade-off (Section 4.1).
- We demonstrate that `Patch Gaussian` can be combined with other regularization strategies (Section 4.2) and data augmentation policies (Section 4.3).
- We perform a frequency-based analysis (Yin et al., 2019) of models trained with `Patch Gaussian` and find that they can better leverage high-frequency information in lower layers, while not being too sensitive to them at later ones (Section 5.1).
- We show a similar method can be used in adversarial training, suggesting under-explored questions about training distributions' effect on out-of-distribution robustness (Section 5.2).

## 2 PRELIMINARIES

We start by considering two data augmentations: `Cutout` (DeVries & Taylor, 2017) and `Gaussian` (Grandvalet & Canu, 1997). The former sets a random patch of the input image to a constant (mean pixel in the dataset) in order to improve clean accuracy. The latter adds independent Gaussian noise to each pixel of the input image, which directly increases robustness to Gaussian noise.

### 2.1 CUTOUT AND GAUSSIAN EXHIBIT A TRADE-OFF BETWEEN ACCURACY AND ROBUSTNESS

We compare the effectiveness of `Gaussian` and `Cutout` data augmentation for accuracy and robustness by measuring the performance of models trained with each on clean as well as corrupted data. Here, robustness is defined as average accuracy of the model, when tested on data corrupted by various $\sigma$ (0.1, 0.2, 0.3, 0.5, 0.8, 1.0) of Gaussian noise, relative to the clean accuracy:

$$\text{Relative Gaussian Robustness} = \mathbb{E}_{\sigma}(\text{Accuracy on Data Corrupted by } \sigma) - \text{Clean Accuracy}$$

Fig. 2 highlights an apparent trade-off in using these methods. In accordance to previous work (De-Vries & Taylor, 2017), `Cutout` improves accuracy on clean test data. Despite this, we find it does not lead to increased robustness. Conversely, training with higher $\sigma$ of `Gaussian` can lead to increased

robustness to Gaussian noise, but also leads to decreased accuracy on clean data. Therefore, any robustness gains are offset by poor overall performance: a model with a perfect Relative Robustness of 0, but whose clean accuracy dropped to $50\%$ will be wrong half the time, even on clean data.

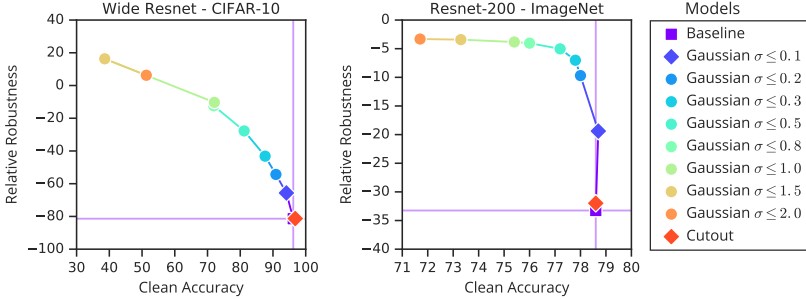

Figure 2: The apparent robustness-accuracy trade-off between `Cutout` and `Gaussian` augmentations. Each dot represents a model trained with different augmentations and hyper-parameters. The y-axis is the change in accuracy when tested on data corrupted with Gaussian noise at various $\sigma$ (average corrupted accuracy minus clean accuracy). The diamond indicates augmentation hyper-parameters selected by the method in Section 3.2.

At first glance, these results seem to reinforce the findings of previous work (Tsipras et al., 2018), indicating that robustness comes at the cost of generalization, which would offset any benefits of improved robustness. In the following sections, we will explore whether there exists augmentation strategies that do not exhibit this limitation.

## 3 METHOD

Each of the two methods seen so far achieves one half of our stated goal: either improving robustness or slightly improving/maintaining clean test accuracy, but never both. To explore whether this observed trade-off is fundamental, we introduce `Patch Gaussian`, a technique that combines the noise robustness of `Gaussian` with the slightly improved clean accuracy of `Cutout`. Our method is intentionally simple but, as we'll see, it's powerful enough to overcome the limitations described and beats complex training schemes designed to provide robustness.

### 3.1 PATCH GAUSSIAN

`Patch Gaussian` works by adding a $W \times W$ patch of Gaussian noise to the image (Figure 3). As with `Cutout`, the center of the patch is sampled to be within the image. By varying the size of this patch and the maximum standard deviation of noise sampled $\sigma_{max}$, we can interpolate between `Gaussian` (which applies additive Gaussian noise to the whole image) and an approximation of `Cutout` (which removes all information inside the patch). See Fig. 9 for more examples.

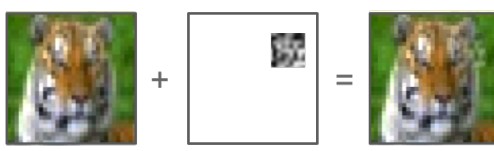

Figure 3: `Patch Gaussian` is the addition of Gaussian noise to pixels in a square patch. It allows us to interpolate between `Gaussian` and `Cutout`, approaching `Gaussian` with increasing patch size and `Cutout` with increasing $\sigma$.

### 3.2 HYPER-PARAMETER SELECTION

Our goal is to learn models that achieve both good clean accuracy and improved robustness to corruptions. Prior work has optimized for one or the other but, as noted before, to meaningfully improve robustness to other distributions, a method can't incur a significant drop in clean accuracy. Therefore, when selecting hyper-parameters, we focus on identifying the models that are most robust while still achieving a minimum accuracy ($Z$) on the clean test data. Values of $Z$ are selected to incur negligible decrease in clean accuracy. As such, they vary per dataset and model, and can be found in the Appendix (Table 5). If no model has clean accuracy $\geq Z$, we report the model with highest clean accuracy, unless otherwise specified.

We find that patch sizes around 25 on CIFAR ($\leq$250 on ImageNet, i.e.: uniformly sampled with maximum value 250) with $\sigma \leq 1.0$ generally perform the best. A complete list of selected hyper-parameters for all augmentations can be found in Table 5.

We are interested in out-of-distribution robustness, and report performance of selected models on Common Corruption (Hendrycks & Dietterich, 2018). However, when selecting hyper-parameters, we use Relative Gaussian Robustness as a stand-in for "robustness." Ford et al. (2019) indicates that this metric is correlated with performance on Common Corruptions, so selected models should be generally robust beyond Gaussian corruptions. By picking models based on robustness to Gaussian noise, we ensure that our selection process does not overfit to the Common Corruptions benchmark.

## 4    RESULTS

Models trained with `Patch Gaussian` overcome the observed trade-off and gain robustness to Gaussian noise while maintaining clean accuracy (Fig. 1). Because Gaussian robustness is only used for hyper-parameter selection, we omit these results, but refer the curious reader to Appendix Fig. 7.

Instead, we report how this Gaussian robustness translates into better Common Corruption robustness, which is in line with reports of the correlation between the two (Ford et al., 2019). In doing so, we establish a new state of the art in the Common Corruptions benchmark (Section 4.1), despite the simplicity of our method when compared with the previous best (Geirhos et al., 2018a). We then show that `Patch Gaussian` can be used in complement to other common regularization strategies (Section 4.2) and data augmentation policies (Cubuk et al., 2018) (Section 4.3).

### 4.1    TRAINING WITH PATCH GAUSSIAN IMPROVES COMMON CORRUPTION ROBUSTNESS

In this section, we look at how our augmentations impact robustness to corruptions beyond Gaussian noise. Rather than focusing on adversarial examples that are worst-case bounded perturbations, we focus on a more general set of corruptions (Gilmer et al., 2018) that models are likely to encounter in real-world settings: the Common Corruptions benchmark (Hendrycks & Dietterich, 2018). This benchmark, also referred to as CIFAR-C and ImageNet-C, is composed of images transformed with 15 corruptions, at 5 severities each. Each is designed to model transformations commonly found in real-world settings, such as brightness, different weather conditions, and different kinds of noise.

Table 1 shows that `Patch Gaussian` achieves state of the art on both of these benchmarks in terms of mean Corruption Error (mCE). A "Corruption Error" is a model's average error over 5 severities of a given corruption, normalized by the same average of a baseline model. However, ImageNet-C was released in compressed JPEG format (ECMA International, 2009), which alters the corruptions applied to the raw pixels. Therefore, we report results on the benchmark as-released ("Original mCE")[1] as well as a version of 12 corruptions without the extra compression ("mCE"). Additionally, because `Patch Gaussian` is a noise-based augmentation, we wanted to verify whether its gains on this benchmark were solely due to improved performance on noise-based corruptions (Gaussian Noise, Shot Noise, and Impulse Noise). To do this, we also measure the models' average performance on all *other* corruptions, reported as "Original mCE (-noise)", and "mCE (-noise)".

The models used to normalize Corruption Errors are the "Baselines" trained with only flip and crop data augmentation. The one exception is Original mCE ImageNet, where we use the AlexNet baseline to be directly comparable with previous work (Hendrycks & Dietterich, 2018; Geirhos et al., 2018a).

On CIFAR, we compare with an adversarially-trained model (Madry et al., 2017). On ImageNet, we compare with a model trained with Random Erasing (Zhong et al., 2017), as well as a shape-biased model "SIN+IN ftIN" (Geirhos et al., 2018a). Finally, previous work (Yin et al., 2019) has found that augmentation diversity is a key component of robustness gains. To confirm that `Patch Gaussian`'s gains aren't simply a result of using multiple augmentations, we also report results on training with `Cutout` and `Gaussian` applied in sequence ("Cutout & Gaussian" and "Gaussian & Cutout"), as well as to 50% of batches ("Gaussian or Cutout").

We observe that `Patch Gaussian` outperforms all models, even on corruptions like fog where `Gaussian` hurts performance (Ford et al., 2019). Scores for each corruption can be found in the

---

[1]We note that, in our implementation, Original mCE used the $299 \times 299$ images provided by the benchmark, which were originally intended for Inception models. Preliminary experiments indicate this does not change our qualitative analysis, but we will update the paper with evaluations on $224 \times 224$ when possible.

Table 1: `Patch Gaussian` achieves state of the art in the CIFAR-C (left) and ImageNet-C (right) robustness benchmarks while maintaining clean test accuracy. "Original mCE" refers to the jpeg-compressed benchmark, as used in Geirhos et al. (2018a); Hendrycks & Dietterich (2018). "mCE" is a version of it without the extra jpeg compression. Note that `Patch Gaussian` improves robustness even in corruptions that aren't noise-based. *`Cutout 16` is presented for direct comparison with DeVries & Taylor (2017); Gastaldi (2017). For Resnet-200, we also present `Gaussian` at a higher $\sigma$ to highlight the accuracy-robustness trade-off. Augmentation hyper-parameters were selected based on the method in Section 3.2 and can be found in Appendix. See text for details.

| | Augmentation | Test Accuracy | mCE | mCE (-noise) |
|---|---|---|---|---|
| Wide Resnet-28-10 | Adversarial | 87.3% | 1.049 | 1.157 |
| | Baseline | 96.2% | 1.000 | 1.000 |
| | Cutout | 96.8% | 1.265 | 1.185 |
| | Cutout 16* | **97.0%** | 1.002 | 0.953 |
| | Gaussian | 94.1% | 0.887 | 0.995 |
| | Patch Gaussian | 96.6% | **0.797** | **0.858** |
| Shake 112 | Baseline | 96.8% | 1.000 | 1.000 |
| | Cutout | 97.1% | 0.946 | 0.930 |
| | Cutout 16* | **97.5%** | 0.912 | 0.872 |
| | Gaussian | 94.6% | 0.977 | 1.111 |
| | Patch Gaussian | 97.2% | **0.713** | **0.776** |

| | Augmentation | Test Accuracy | Original mCE | Original mCE (-noise) | mCE | mCE (-noise) |
|---|---|---|---|---|---|---|
| Resnet-50 | SIN+IN ftIN | 76.7% | 0.738 | 0.731 | - | - |
| | Random Erasing | 76.1% | 0.785 | 0.795 | 1.035 | 1.037 |
| | Baseline | **76.4%** | 0.753 | 0.763 | 1.00 | 1.00 |
| | Cutout | 76.2% | 0.758 | 0.766 | 1.007 | 1.005 |
| | Gaussian | 75.6% | 0.739 | 0.754 | 0.898 | 0.991 |
| | Patch Gaussian | 76.0% | **0.714** | **0.736** | **0.872** | **0.969** |
| | Cutout & Gaussian | 75.8% | 0.757 | 0.769 | 0.934 | 1.015 |
| | Gaussian & Cutout | 75.8% | 0.761 | 0.770 | 0.923 | 1.013 |
| | Gaussian or Cutout | 76.0% | 0.753 | 0.765 | 0.926 | 1.009 |
| Resnet-200 | Baseline | 78.6% | 0.675 | 0.686 | 0.881 | 0.883 |
| | Cutout | 78.6% | 0.671 | 0.687 | 0.874 | 0.884 |
| | Gaussian | **78.7%** | 0.658 | 0.678 | 0.795 | 0.881 |
| | Gaussian ($\sigma \leq 0.2$) | 78.1% | 0.644 | 0.665 | 0.784 | 0.874 |
| | Patch Gaussian | **78.7%** | **0.604** | **0.634** | **0.736** | **0.818** |

Appendix (Tables 6&7). We observe small gains on low capacity models (ResNet-50 & Wide ResNet), but find large robustness improvements with higher capacity ones (ResNet-200 & Shake 112).

These results are surprising: achieving robustness on par with the human visual system is thought to require major changes in training procedures and datasets. Training shape-biased models (Geirhos et al., 2018a) involves creating a custom dataset of style-transferred images, which is a computationally-expensive operation. Even with these, the most robust model reported SIN+IN displays a significant drop in clean accuracy. Because of this, our main comparison is with SIN+IN ftIN, which is fine-tuned on ImageNet. A comparison with SIN+IN can be found in Appendix Table 8.

In sum, despite its simplicity, `Patch Gaussian` achieves a substantial decrease in mCE relative to other models, indicating that current methods have not reached the theoretical trade-off (Tsipras et al., 2018), and that complex training schemes (Geirhos et al., 2018a) are not needed for robustness.

## 4.2 PATCH GAUSSIAN CAN BE COMBINED WITH OTHER REGULARIZATION STRATEGIES

Since `Patch Gaussian` has a regularization effect on the models trained above, we compare it with other regularization methods: larger weight decay, label smoothing, and dropblock (Table 2). We find that while label smoothing improves clean accuracy, it weakens the robustness in all corruption metrics we have considered. This agrees with the theoretical prediction from Cubuk et al. (2017), which argued that increasing the confidence of models would improve robustness, whereas label smoothing reduces the confidence of predictions. We find that increasing the weight decay from the default value used in all models does not improve clean accuracy or robustness.

Here, we focus on analyzing the interaction of different regularization methods with `Patch Gaussian`. Previous work indicates that improvements on the clean accuracy appear after training with Dropblock for 270 epochs (Ghiasi et al., 2018), but we did not find that training for 270 epochs changed our analysis. Thus, we present models trained at 90 epochs for direct comparison with other results. Due to the shorter training time, Dropblock does not improve clean accuracy, yet it does make the model more robust (relative to baseline) according to all corruption metrics we consider.

We find that using label smoothing in addition to `Patch Gaussian` has a mixed effect, it improves clean accuracy while slightly improving robustness metrics except for the Original mCE. Combining Dropblock with `Patch Gaussian` reduces the clean accuracy relative to the `Patch Gaussian`-only model, as Dropblock seems to be a strong regularizer when used for 90 epochs. However, using

Table 2: `Patch Gaussian` can be used with other regularization methods for improved robustness. "Original mCE" refers to the jpeg-compressed benchmark, as used in Geirhos et al. (2018a); Hendrycks & Dietterich (2018). "mCE" is a version of it without the extra jpeg compression. All of the models are ResNet-50 trained on ImageNet with same hyperparameters for 90 epochs.

| | Regularization | Test Accuracy | Original mCE | Original mCE (-noise) | mCE | mCE (-noise) |
|---|---|---|---|---|---|---|
| Resnet-50 | Label Smoothing | **76.7%** | 0.760 | 0.765 | 1.01 | 1.01 |
| | Larger Weight Decay (0.001) | 74.9% | 0.766 | 0.777 | 1.02 | 1.03 |
| | Dropblock | 76.3% | 0.734 | 0.743 | 0.971 | 0.974 |
| | Patch Gaussian + Label Smoothing | 76.5% | 0.720 | 0.734 | **0.868** | 0.966 |
| | Patch Gaussian + Dropblock | 75.7% | **0.708** | **0.726** | 0.870 | **0.961** |

Dropblock and `Patch Gaussian` together leads to the best robustness performance. These results indicate that `Patch Gaussian` can be used in conjunction with existing regularization strategies.

### 4.3 PATCH GAUSSIAN CAN BE COMBINED WITH AUTOAUGMENT FOR IMPROVED RESULTS

Knowing that `Patch Gaussian` can be combined with other regularizers, it's natural to ask whether it can also be combined with other data augmentation policies. Previous work has found that varied augmentation policies have a large positive impact on model robustness (Yin et al., 2019). In this section, we verify that `Patch Gaussian` can be added to these policies for further gains.

Table 3 highlights models trained with AutoAugment (Cubuk et al., 2018). For fair comparison of mCE scores, we train all models with the best AutoAugment policy, but without contrast and Inception color pre-processing, as those are present in the Common Corruptions benchmark. This process is imperfect since AutoAugment has *many* operations, some of which could be correlated with corruptions. Regardless, we find that `Patch Gaussian` improves robustness over AutoAugment.

Because AutoAugment leads to state of the art accuracies, we are interested in seeing how far it can be combined with `Patch Gaussian` to improve results. Therefore, and unlike previous experiments, models are trained for 180 epochs to yield best results possible.

Table 3: `Patch Gaussian` can be combined with AutoAugment (Cubuk et al., 2018) data augmentation policy for improved results. "Original mCE" refers to the jpeg-compressed benchmark, as used in Geirhos et al. (2018a); Hendrycks & Dietterich (2018). "mCE" is a version of it without the extra jpeg compression. All of the models are ResNet-50 trained on ImageNet with best AutoAugment policy for 180 epochs, to highlight improvements.

| | Model (trained with AutoAugment) | Test Accuracy | Original mCE | Original mCE (-noise) | mCE | mCE (-noise) |
|---|---|---|---|---|---|---|
| Res Net 50 | Baseline | 77.0% | 0.674 | 0.697 | 0.855 | 0.882 |
| | Patch Gaussian ($W$=150, $\sigma \le 0.5$) | **77.3%** | **0.656** | **0.682** | **0.779** | **0.863** |

## 5 DISCUSSION

In an attempt to understand `Patch Gaussian`'s performance, we perform a frequency-based analysis of models trained with various augmentations using the method introduced in Yin et al. (2019).

First, we perturb each image in the dataset with noise sampled at each orientation and frequency in Fourier space. Then, we measure changes in the network activations and test error when evaluated with these Fourier-noise-corrupted images: we measure the change in $\ell_2$ norm of the tensor directly after the first convolution, as well as the absolute test error. This procedure yields a heatmap, which indicates model sensitivity to different frequency and orientation perturbations in the Fourier domain. Each image in Fig 4 shows first layer (or test error) sensitivity as a function of frequency and orientation of the sampled noise, with the middle of the image containing the lowest frequencies, and the edges of the image containing highest frequencies.

For CIFAR-10 models, we present this analysis for the entire Fourier domain, with noise sampled with norm 4. For ImageNet, we focus our analysis on lower frequencies that are more visually salient add noise with norm 15.7.

Note that for `Cutout` and `Gaussian`, we chose larger patch sizes and $\sigma$s than those selected with the method in Section 3.2 in order to highlight the effect of these augmentations on sensitivity. Heatmaps of other models can be found in the Appendix (Figure 11).

### 5.1 FREQUENCY-BASED ANALYSIS OF MODELS TRAINED WITH PATCH GAUSSIAN

We confirm findings by Yin et al. (2019) that `Gaussian` encourages the model to learn a low-pass filter of the inputs. Models trained with this augmentation, then, have low test error sensitivity at high frequencies, which could help robustness. However, valuable high-frequency information (Brendel & Bethge, 2019) is being thrown out at low layers, which could explain the lower test accuracy.

We further find that `Cutout` encourages the use of high-frequency information, which could help explain its improved generalization performance. Yet, it does not encourage lower test error sensitivity, which explains why it doesn't improve robustness either.

`Patch Gaussian`, on the other hand, seems to allow high-frequency information through at lower layers, but still encourages relatively lower test error sensitivity at high frequencies. Indeed, when we measure accuracy on images filtered with a high-pass filter, we see that `Patch Gaussian` models can maintain accuracy in a similar way to the baseline and to `Cutout`, where `Gaussian` fails to. See Figure 4 for full results.

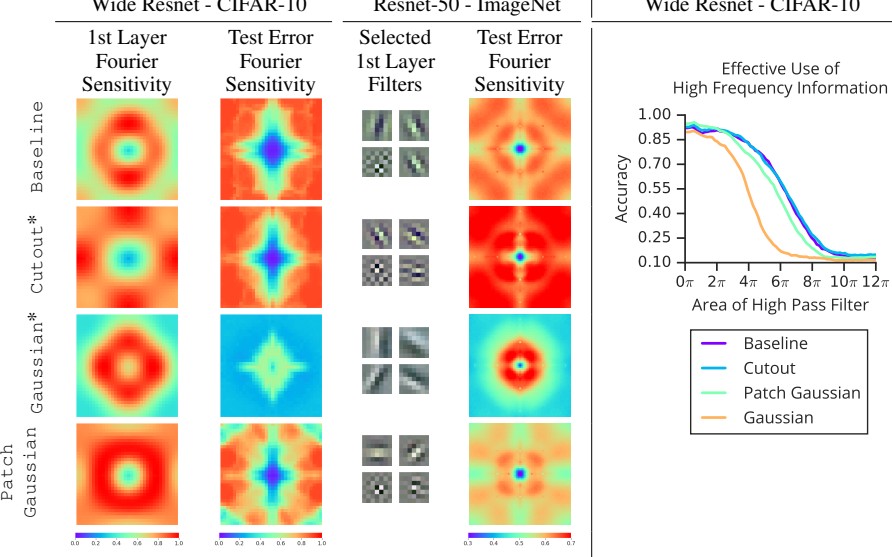

Figure 4: (left) Fourier analysis of various models, using method from Yin et al. (2019). Heatmaps depict model sensitivity to various sinusoidal gratings. `Cutout` encourages the use of high frequencies in earlier layers, but its test error remains too sensitive to them. `Gaussian` learns low-pass filtering of features, which increases robustness at later layers, but makes lower layers too invariant to high-frequency information (thus hurting accuracy). `Patch Gaussian` allows high frequencies to be used in lower layers, and its test error remains relatively robust to them. This can also be seen by the presence of high-frequency kernels in the first layer filters of the models (or lack thereof, in the case of `Gaussian`). (right) Indeed, `Patch Gaussian` models match the performance of `Cutout` and `Baseline` when presented with only the high frequency information of images, whereas `Gaussian` fails to effectively utilize this information (see Appendix Fig. 12 for experiment details). This pattern of reduced sensitivity of predictions to high frequencies in the input occurs across all augmentation magnitudes, but here we use larger patch sizes and $\sigma$ of noise to highlight the differences in models indicated by *. See text for details.

Understanding the impact of data distributions and noise on representations has been well-studied in neuroscience (Barlow et al., 1961; Simoncelli & Olshausen, 2001; Karklin & Simoncelli, 2011). The data augmentations that we propose here alter the distribution of inputs that the network sees, and thus are expected to alter the kinds of representations that are learned. Prior work on efficient coding (Karklin & Simoncelli, 2011) and autoencoders (Poole et al., 2014) has shown how filter properties

change with noise in the unsupervised setting, resulting in lower-frequency filters with `Gaussian`, as we observe in Fig. 4. Consistent with prior work on natural image statistics (Torralba & Oliva, 2003), we find that networks are least sensitive to low frequency noise where spectral density is largest. Performance drops at higher frequencies when the amount of noise we add grows relative to typical spectral density observed at these frequencies. In future work, we hope to better understand the relationship between naturally occurring properties of images and sensitivity, and investigate whether training with more naturalistic noise can yield similar gains in corruption robustness.

## 5.2 PATCHING ADVERSARIAL TRAINING ALSO LEADS TO IMPROVED ROBUSTNESS

Our results indicate that patching a transformation can prevent overfitting to that particular transformation and maintain clean accuracy. To further confirm this, we train a model with adversarial training applied only to a patch of the training input. Adversarial training is a method of achieving robustness to worst-case perturbations. Models trained in this setting notoriously exhibit decreased clean accuracy, so it is a good candidate to verify whether our robustness gains come from patching.

We train our models with PGD, in a setting equivalent to Madry et al. (2017). For `Patch PGD`, the adversarial perturbation is calculated on the whole image for all steps, and patched after the fact. We also tried calculating PGD on a patch only and found similar results. We select hyper-parameters based on PGD performance on validation set, while maintaining accuracy above 90%. However, in this section we are not interested in improving adversarial robustness performance, but on seeing its effect on robustness to Common Corruptions, to evaluate out-of-distribution (o.o.d.) robustness. We leave an analysis of the effect of patching on adversarial robustness to future work.

Indeed, Table 4 shows that training with `Patch PGD` obtains similar PGD accuracy to training with `PGD`, but maintains most of the clean accuracy of the baseline model. Surprisingly, `Patch PGD` also improves robustness to unseen Common Corruptions, when compared to the baseline without adversarial training, indicating that patching is a generally powerful tool. This also suggests there are unexplored questions regarding the training distribution and how that translates into i.i.d and o.o.d generalization. We hope to explore these in future work.

Table 4: Patching also helps models trained adversarially to maintain clean accuracy and gain Common Corruption robustness. All are Wide-ResNet models trained on CIFAR with PGD with eps 8 for 7 steps, with step size 2), just like in Madry et al. (2017).

|  | Test Accuracy | PGD Accuracy | mCE | mCE (-noise) |
|---|---|---|---|---|
| Baseline | **96.2%** | 00.3% | 1.000 | 1.000 |
| PGD | 85.7% | 50.0% | 1.390 | 1.581 |
| Patch PGD ($W$=19) | 93.5% | **50.8%** | **0.814** | **0.894** |

## 6 CONCLUSION

In this work, we introduced a simple data augmentation operation, `Patch Gaussian`, which improves robustness to common corruptions without incurring a drop in clean accuracy. For models that are large relative to the dataset size (like ResNet-200 on ImageNet and all models on CIFAR-10), `Patch Gaussian` improves clean accuracy and robustness concurrently. We showed that `Patch Gaussian` achieves this by interpolating between two standard data augmentation operations `Cutout` and `Gaussian`. Finally, we analyzed the sensitivity to noise in different frequencies of models trained with `Cutout` and `Gaussian`, and showed that `Patch Gaussian` combines their strengths without inheriting their weaknesses. Our method is much simpler than previous state of the art, and can be used in conjunction with other regularization and data augmentation strategies, indicating it is generally useful. We end by showing that applying perturbations in patches can be a powerful method to vary training distributions in the adversarial setting. Our results indicate current methods have not reached a fundamental robustness/accuracy trade-off, and that future work is needed to understand the effect of training distributions in o.o.d. robustness.

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

APPENDIX

GAUSSIAN ROBUSTNESS

Fig. 5 shows the accuracy/robustness trade-off of models trained with various hyper-parameters of Cutout and Gaussian. Fig. 6 shows clean accuracy change of models trained with various hyper-parameters of Patch Gaussian. Fig. 7 shows how Patch Gaussian can overcome the observed trade-off and gain Gaussian robustness in various models and datasets.

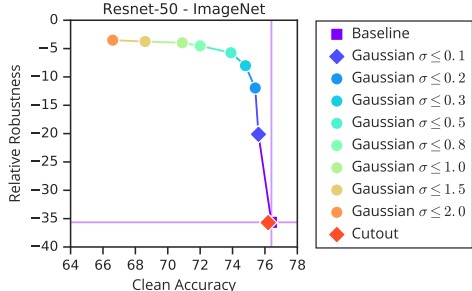

Figure 5: Accuracy/robustness trade-off observed for Cutout and Gaussian on Resnet-50 models. See Figure 2 for details.

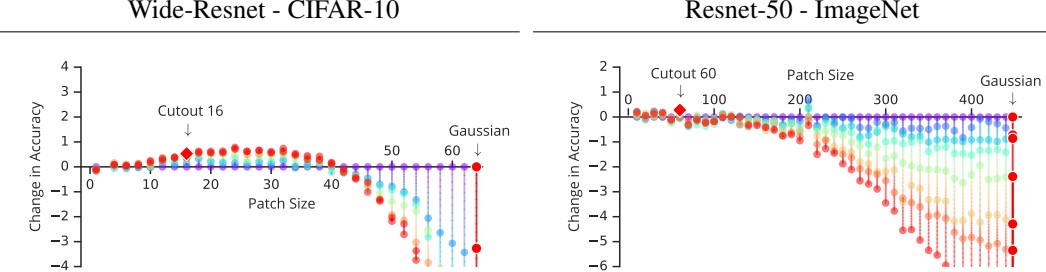

Figure 6: Patch Gaussian hyper-parameter sweep for Wide-Resnet on CIFAR-10 (left) and RN50 on Imagenet (right). Patch Gaussian approaches Gaussian with increasing patch size and Cutout with increasing $\sigma$. Each dot is a model trained with different hyper parameters. Colors indicate different $\sigma$.

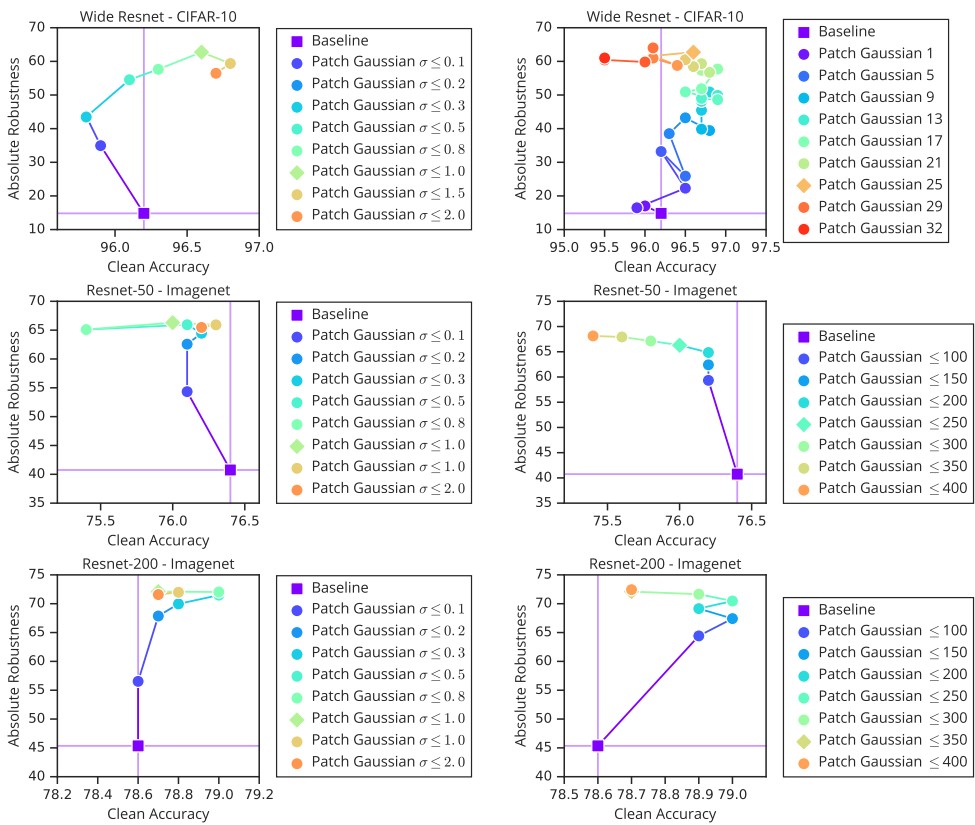

Figure 7: Training with `Patch Gaussian` improves clean data accuracy and Gaussian robustness simultaneously. Each dot represents a model trained with various $\sigma$ (left) or patch sizes (right), while keeping the other fixed at the value indicated by the diamond. The y-axis is the average absolute accuracy when tested on data corrupted by Gaussian noise at various $\sigma$. The diamond indicates the augmentation hyper-parameters selected by the method in Section 3.2.

MODELS, DATASETS, & IMPLEMENTATION DETAILS

We run our experiments on CIFAR-10 (Krizhevsky & Hinton, 2009) and ImageNet (Deng et al., 2009) datasets. On CIFAR-10, we use the Wide-ResNet-28-10 model (Zagoruyko & Komodakis, 2016), as well as the Shake-shake-112 model (Gastaldi, 2017), trained for 200 epochs and 600 epochs respectively. The Wide-ResNet model uses a initial learning rate of 0.1 with a cosine decay schedule. Weight decay is set to be $5 \times 10^{-4}$ and batch size is 128. We train all models, including the baseline, with standard data augmentation of horizontal flips and pad-and-crop. Our code uses the same hyper parameters as Cubuk et al. (2018)

On ImageNet, we use the ResNet-50 and Resnet-200 models (He et al., 2016), trained for 90 epochs. We use a weight decay rate of $1 \times 10^{-4}$, global batch size of 512 and learning rate of 0.2. The learning rate is decayed by 10 at epochs 30, 60, and 80. We use standard data augmentation of horizontal flips and crops. All CIFAR-10 and ImageNet experiments use the listed hyper-parameters above, unless specified otherwise.

To apply Gaussian, we uniformly sample a standard deviation $\sigma$ from 0 up to some maximum value $\sigma_{max}$, and add i.i.d. noise sampled from $\mathcal{N}(0, \sigma^2)$ to each pixel. To apply Cutout, we use a fixed patch size $W$, and randomly set a square region with size $W \times W$ to the constant mean of each RGB channel in the dataset. As in DeVries & Taylor (2017), the patch location is randomly sampled and can lie outside of the $32 \times 32$ CIFAR-10 (or $224 \times 224$ ImageNet) image but its center is constrained to lie within it. Patch sizes and $\sigma_{max}$ are selected based on the method in Section 3.2.

All image transformations, including Patch Gaussian, are performed on images with unnormalized pixel values in $[0, 1]$ range. For all images, standard random flipping and cropping are applied immediately *after* any augmentations mentioned on CIFAR-10 (*before*, on ImageNet). After noise-based augmentations, images are clipped to the $[0, 1]$ range. Augmentation hyper-parameters are selected based on the method in Sec. 3.2 and shown in Tab. 5.

Table 5: Augmentation hyper-parameters selected with the method in Section 3.2 for each model/dataset. *Indicates manually-chosen stronger hyper-parameters, used to highlight the effect of the augmentation on the models. "$\leq$" indicates that the value is uniformly sampled up to the given maximum value.

| | | $Z$ | Augmentation | $W$ | $\sigma$ | Other |
|---|---|---|---|---|---|---|
| CIFAR-10 | Wide Resnet-28-10 | 96.5% | Cutout | $= 12$ | - | |
| | | | Gaussian | - | $\leq 0.1$ | |
| | | | Patch Gaussian | $= 25$ | $\leq 1.0$ | |
| | | | Cutout* | $= 22$ | - | |
| | | | Gaussian* | - | $\leq 1.0$ | |
| | Shake 112 | 97.0% | Cutout | $= 7$ | - | |
| | | | Gaussian | - | $\leq 0.1$ | |
| | | | Patch Gaussian | $= 26$ | $\leq 1.0$ | |
| ImageNet | Resnet-50 | 76.0% | Random Erasing | $= 120$ | - | |
| | | | Baseline | - | - | includes weight decay $= 0.0001$ |
| | | | Cutout | $= 60$ | - | |
| | | | Gaussian | - | $\leq 0.1$ | |
| | | | Patch Gaussian | $\leq 250$ | $\leq 1.0$ | |
| | | | Cutout* | $= 200$ | - | |
| | | | Gaussian* | - | $\leq 1.0$ | |
| | | | Larger Weight Decay | - | - | 0.001 |
| | | | Dropblock | - | - | groups $= 3,4$; keep prob $= 0.9$ |
| | | | Label Smoothing | - | - | 0.1 |
| | Resnet 200 | 78.5% | Baseline | - | - | includes weight decay $= 0.0001$ |
| | | | Cutout | $= 30$ | - | |
| | | | Gaussian | - | $\leq 0.1$ | |
| | | | Patch Gaussian | $\leq 350$ | $\leq 1.0$ | |

PATCH GAUSSIAN

Fig. 8 shows an implementation of `Patch Gaussian`, highlighting its simplicity. Fig. 9 shows the effect of applying `Patch Gaussian` at various settings to an ImageNet image. Figs. 6 and 7 show full Original Corruption Errors and Corruption Errors for ImageNet models. Fig. 8 shows a comparison of `Patch Gaussian` with SIN+IN.

```python
def _get_patch_mask(patch_size):
  # randomly sample location in the image
  x = tf.random.uniform([], minval=0, maxval=224, dtype=tf.int32)
  y = tf.random.uniform([], minval=0, maxval=224, dtype=tf.int32)
  x, y = tf.cast(x, tf.float32), tf.cast(y, tf.float32)

  # compute where the patch will start and end
  startx, starty = x - tf.floor(patch_size/2), y - tf.floor(patch_size/2)
  endx, endy = x + tf.ceil(patch_size/2), y + tf.ceil(patch_size/2)
  startx, starty = tf.maximum(startx, 0), tf.maximum(starty, 0)
  endx, endy = tf.minimum(endx, 224), tf.minimum(endy, 224)

  # now let's convert these into how much we need to pad the patch
  lower_pad, upper_pad = 224 - endy, starty
  left_pad, right_pad = startx, 224 - endx
  padding_dims = [[upper_pad, lower_pad], [left_pad, right_pad]]

  # create mask
  mask = tf.pad(tf.zeros([endy - starty, endx - startx]),
                padding_dims, constant_values=1)
  mask = tf.expand_dims(mask, -1)
  mask = tf.tile(mask, [1, 1, 3])
  return tf.equal(mask, 0)

def patch_gaussian(image, patch_size, max_scale, sample_up_to):
  """Returns image with Patch Gaussian applied."""

  if sample_up_to:
    patch_size = tf.random.uniform([], 1, patch_size, tf.int32)
    # otherwise, patch_size is fixed.

  # make image (which is [0, 255]) be [0, 1]
  image = image / 255.0

  # uniformly sample scale from 0 to given scale
  scale = max_scale * tf.random.uniform([], minval=0, maxval=1)

  # apply gaussian to copy of image. Will be used to replace patch in image
  gaussian = tf.random.normal(stddev=scale, shape=image.shape)
  image_plus_gaussian = tf.clip_by_value(image + gaussian, 0, 1)

  # create mask and apply patch
  image = tf.where(_get_patch_mask(patch_size),
                   image_plus_gaussian, image)

  # scale back to [0, 255]
  return image * 255
```

Figure 8: TensorFlow implementation of `Patch Gaussian`

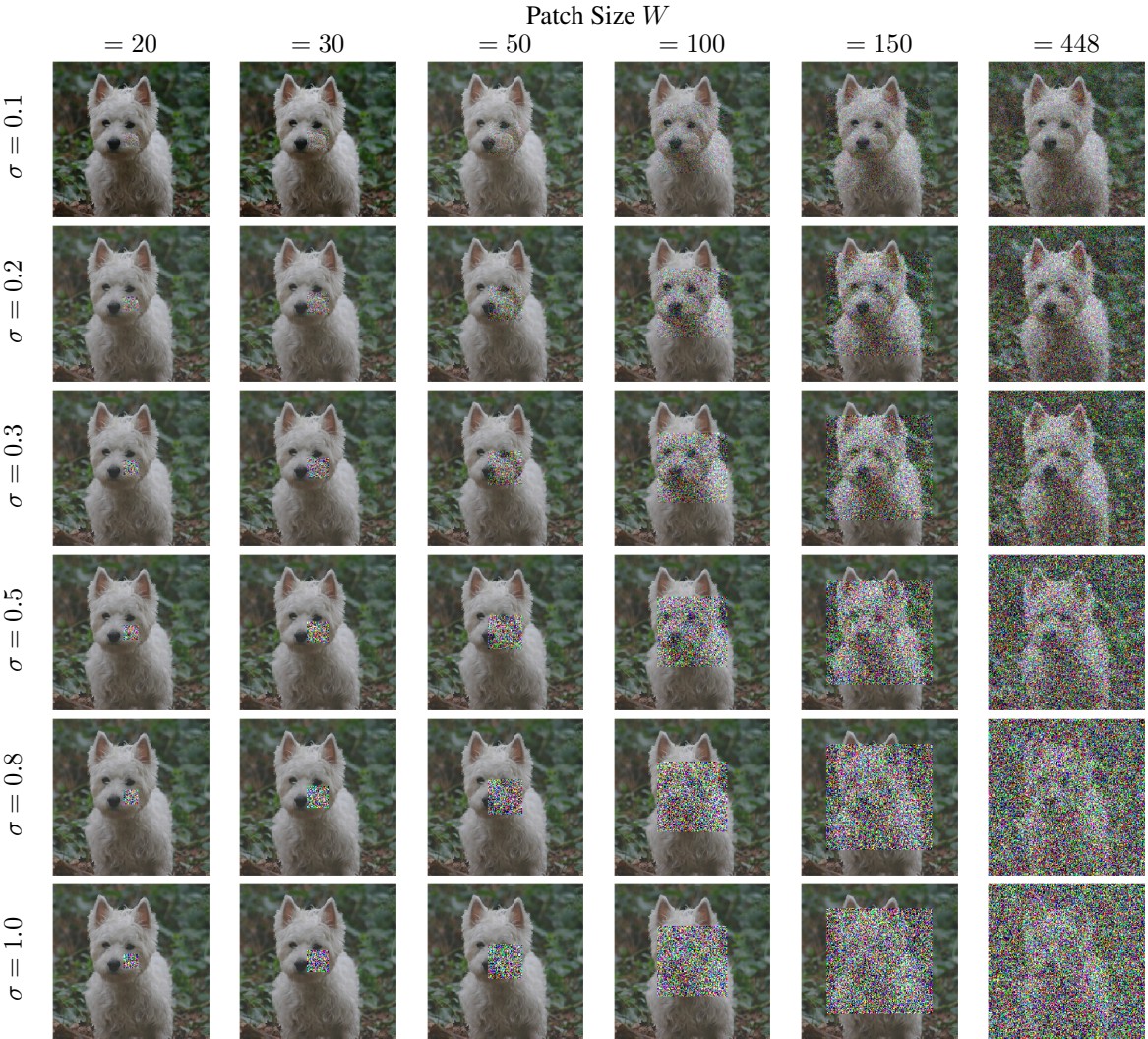

Figure 9: Images modified with `Patch Gaussian`, with centered patch, at various $W$ & $\sigma$.

Table 6: Full original corruption errors (Original CEs) for ImageNet models trained with different augmentation strategies.

| | Augmentation | Noise | | | Blur | | | |
|---|---|---|---|---|---|---|---|---|
| | | Gaussian | Shot | Impulse | Defocus | Glass | Motion | Zoom |
| Resnet-50 | Baseline | 0.705 | 0.722 | 0.716 | 0.815 | 0.915 | 0.810 | 0.817 |
| | Cutout | 0.720 | 0.727 | 0.720 | 0.798 | 0.923 | 0.821 | 0.813 |
| | Gaussian | 0.677 | 0.681 | 0.677 | 0.781 | **0.864** | 0.813 | 0.808 |
| | Patch Gaussian | **0.623** | **0.633** | **0.624** | **0.751** | 0.898 | **0.782** | **0.783** |
| Resnet-200 | Baseline | 0.622 | 0.641 | 0.629 | 0.735 | 0.867 | 0.722 | 0.739 |
| | Cutout | 0.594 | 0.619 | 0.600 | 0.714 | 0.870 | 0.713 | 0.737 |
| | Gaussian | 0.573 | 0.583 | 0.575 | 0.723 | 0.814 | 0.737 | 0.741 |
| | Patch Gaussian | **0.486** | **0.498** | **0.478** | **0.649** | **0.805** | **0.693** | **0.687** |

| | Augmentation | Weather | | | | Digital | | | |
|---|---|---|---|---|---|---|---|---|---|
| | | Snow | Frost | Fog | Bright | Contrast | Elastic | Pixel | JPEG |
| Resnet-50 | Baseline | 0.827 | 0.756 | 0.589 | **0.582** | 0.748 | 0.753 | 0.799 | 0.747 |
| | Cutout | 0.839 | 0.764 | 0.599 | 0.586 | 0.747 | 0.752 | 0.803 | 0.752 |
| | Gaussian | 0.821 | **0.726** | 0.597 | 0.592 | 0.754 | **0.720** | 0.805 | 0.763 |
| | Patch Gaussian | **0.806** | 0.739 | **0.566** | 0.592 | **0.714** | 0.736 | **0.743** | **0.722** |
| Resnet-200 | Baseline | 0.754 | 0.694 | 0.497 | 0.520 | 0.658 | 0.669 | 0.696 | 0.681 |
| | Cutout | 0.741 | 0.684 | 0.507 | 0.516 | 0.671 | 0.670 | 0.751 | 0.672 |
| | Gaussian | 0.731 | 0.653 | 0.525 | 0.514 | 0.699 | 0.641 | 0.693 | 0.660 |
| | Patch Gaussian | **0.697** | **0.633** | **0.476** | **0.506** | **0.627** | **0.625** | **0.613** | **0.593** |

Table 7: Full corruption errors (CEs) for ImageNet models trained with different augmentation strategies.

| | Augmentation | Noise | | | Blur | |
|---|---|---|---|---|---|---|
| | | Gaussian | Shot | Impulse | Defocus | Zoom |
| Resnet-50 | Baseline | 1.000 | 1.000 | 1.000 | 1.000 | 1.000 |
| | Cutout | 1.015 | 1.013 | 1.008 | 0.979 | 1.000 |
| | Gaussian | 0.620 | 0.625 | 0.618 | 0.950 | 0.999 |
| | Patch Gaussian | **0.585** | **0.577** | **0.577** | **0.922** | **0.963** |
| Resnet-200 | Baseline | 0.872 | 0.883 | 0.864 | 0.880 | 0.896 |
| | Cutout | 0.841 | 0.862 | 0.833 | 0.866 | 0.892 |
| | Gaussian | 0.533 | 0.538 | 0.538 | 0.855 | 0.910 |
| | Patch Gaussian | **0.490** | **0.488** | **0.490** | **0.767** | **0.820** |

| | Augmentation | Weather | | | Digital | | | |
|---|---|---|---|---|---|---|---|---|
| | | Frost | Fog | Bright | Contrast | Elastic | Pixel | JPEG |
| Resnet-50 | Baseline | 1.000 | 1.000 | 1.000 | 1.000 | 1.000 | 1.000 | 1.000 |
| | Cutout | 1.005 | 1.017 | 0.991 | 1.008 | 1.009 | 1.026 | 1.011 |
| | Gaussian | **0.919** | 1.073 | 1.019 | 1.051 | **0.967** | 0.974 | **0.966** |
| | Patch Gaussian | 0.976 | **0.978** | **0.990** | **0.956** | 0.982 | **0.957** | 0.998 |
| Resnet-200 | Baseline | 0.912 | 0.862 | 0.888 | 0.861 | 0.882 | 0.848 | 0.922 |
| | Cutout | 0.915 | 0.868 | 0.877 | 0.877 | 0.871 | 0.875 | 0.911 |
| | Gaussian | 0.830 | 0.948 | 0.889 | 0.960 | 0.848 | 0.836 | 0.855 |
| | Patch Gaussian | **0.818** | **0.851** | **0.862** | **0.832** | **0.812** | **0.765** | **0.835** |

Table 8: Comparison with SIN+IN (Geirhos et al., 2018a). By using $Z$=74.6%, `Patch Gaussian` can match SIN+IN's og mCE and test accuracy. Understandably, however, our gains are more concentrated in noise-based corruptions, whereas shape-biased models get gains in other corruptions.

| | Augmentation | Test Accuracy | Original mCE | Original mCE (-noise) |
|---|---|---|---|---|
| Resnet-50 | SIN+IN | 74.6% | **0.693** | **0.699** |
| | Patch Gaussian ($W \leq 400$, $\sigma \leq 0.8$) | **75.6%** | **0.693** | 0.718 |

PATCH GAUSSIAN IMPROVES PERFORMANCE IN OBJECT DETECTION

Since `Patch Gaussian` can be combined with both regularization strategies as well as data augmentation policies, we want to see if it is generally useful beyond classification tasks. We train a RetinaNet detector (Lin et al., 2017) with ResNet-50 backbone (He et al., 2016) on the COCO dataset (Lin et al., 2014). Images for both baseline and `Patch Gaussian` models are horizontally flipped half of the time, after being resized to $640 \times 640$. We train both models for 150 epochs using a learning rate of 0.08 and a weight decay of $1 \times 10^{-4}$. The focal loss parameters are set to be $\alpha = 0.25$ and $\gamma = 1.5$.

Despite being designed for classification, `Patch Gaussian` improves detection performance according to all metrics when tested on the clean COCO validation set (Table 9). On the primary COCO metric mean average precision (mAP), the model trained with `Patch Gaussian` achieves a 1% higher accuracy over the baseline, whereas the model trained with `Gaussian` suffers a 2.9% loss.

Table 9: Mean average precision (mAP) on COCO with baseline augmentation of horizontal flips and `Patch Gaussian`. $mAP_S$, $mAP_M$, and $mAP_L$ refer to mAP for small, medium, and large objects, respectively. $mAP_{50}$ and $mAP_{75}$ refer to mAP at intersection over union values of 50 and 75, respectively. mAP in the final column is the averaged mAP over IoU=0.5:0.05:0.95.

| Tested on | | $mAP_S$ | $mAP_M$ | $mAP_L$ | $mAP_{50}$ | $mAP_{75}$ | mAP |
|---|---|---|---|---|---|---|---|
| Clean Data | Baseline | 15.6 | 36.9 | 48.3 | 50.8 | 35.6 | 33.2 |
| | Gaussian ($\sigma \leq 1.0$) | 13.1 | 32.6 | 44.0 | 45.7 | 31.2 | 29.3 |
| | Patch Gaussian ($W$=200, $\sigma \leq 1.0$) | **16.1** | **37.9** | **50.3** | **51.9** | **36.5** | **34.2** |
| Gaussian Noise (=0.25) | Baseline | 4.5 | 12.7 | 17.6 | 19.3 | 11.7 | 11.6 |
| | Gaussian ($\sigma \leq 1.0$) | 9.9 | 28.1 | **41.0** | **41.7** | 26.8 | **26.1** |
| | Patch Gaussian ($W$=200, $\sigma \leq 1.0$) | **10.1** | **28.2** | 40.4 | 41.3 | **27.2** | **26.1** |

Next, we evaluate these models on the validation set corrupted by i.i.d. Gaussian noise, with $\sigma = 0.25$. We find that model trained with `Gaussian` and `Patch Gaussian` achieve the highest mAP of 26.1% on the corrupted data, whereas the baseline achieves 11.6%. It is interesting to note that `Patch Gaussian` model achieves a better result on the harder metrics of small object detection and stricter intersection over union (IoU) thresholds, whereas the `Gaussian` model achieves a better result on the easier tasks of large object detection and less strict IOU threshold metric.

Overall, as was observed for the classification tasks, training object detection models with `Patch Gaussian` leads to significantly more robust models without sacrificing clean accuracy.

FOURIER ANALYSIS

Fig. 10 shows a fourier analysis of selected models reported. Fig. 11 shows complete filters for ResNet-50 models. Fig. 12 shows high-pass filters used in high-pass experiment in Fig. 4.

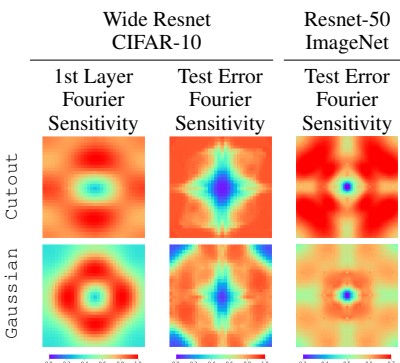

Figure 10: Fourier analysis for `Cutout` and `Gaussian` models selected by the method in Section 3.2. See Figure 4 for details.

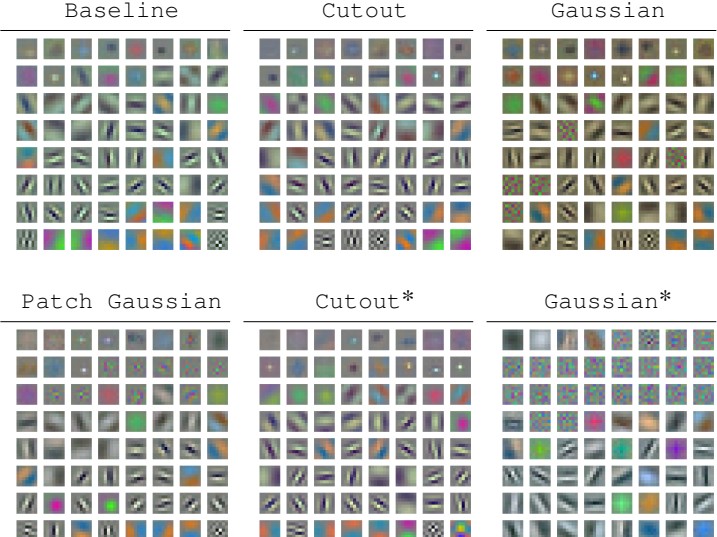

Figure 11: Complete filters for Resnet-50 models trained on ImageNet. * Indicates augmentations with larger patch sizes and $\sigma$. See Figure 4 for details. We again note the presence of filters of high fourier frequency in models trained with `Cutout`* and `Patch Gaussian`. We also note that `Gaussian`* exhibits high variance filters. We posit these have not been trained and have little importance, given the low sensitivity of this model to high frequencies. Future work will investigate the importance of filters on sensitivity.

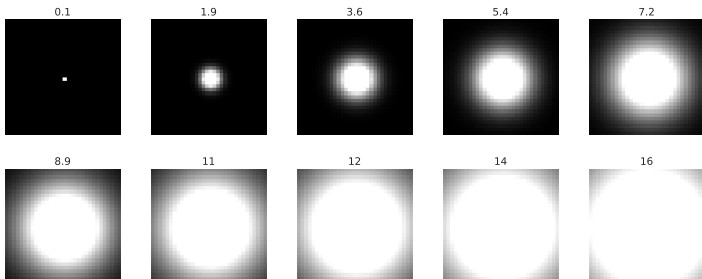

Figure 12: Examples of high pass filters at various radii, in fourier space centered at the zero-frequency component, used in the high-pass experiment of Figure 4.

NOTE ON PGD COMMON CORRUPTION ROBUSTNESS

Ford et al. (2019) reports that `PGD` training helps with corruption robustness. However, they fail to report mCE values for their models. We find that, indeed, `PGD` helps with some corruptions, and when all corruption severities' errors are averaged, it mostly maintains performance (23.8% error, compared to baseline error of 23.51%). However, as table 4 shows, when we properly calculate mCE by normalizing with a baseline model, `PGD` displays much worse robustness, while `Patch PGD` improves performance.

ABLATION STUDY OF FREQUENCY-BASED ANALYSIS

Figure 13 shows the frequency-based analysis (Yin et al., 2019) for models with different hyper-parameters of `Patch Gaussian`.

First, for hyper-parameters $W$=16, $\sigma$=1.0 (center), the reader will note that these are very similar to the frequency sensitivity reported in Figure 4. The main difference being that the smaller patch size (16 vs 25 in Figure 4) makes the model slightly more sensitive to high frequencies. This makes sense since smaller patch size moves the model further away from a `Gaussian`-trained one.

When we make the scale smaller ($W$=16, $\sigma$=0.3, left), less information is corrupted in the patch, which moves the model farther from the one trained with `Cutout` (and therefore closer to a `Gaussian`-trained one). This can be seen in the increased invariance to high frequencies at the first layer, which is reflected in invariance at test error as well.

If we, instead, make the scale larger($W$=16, $\sigma$=2.0, right), we move the model closer to the one trained with `Cutout`. Notice the higher intensity red in the first layer plot, indicating higher sensitivity to high-frequency features. We also see this sensitivity reflected in the test error, which matches the behavior for `Cutout`-trained models.

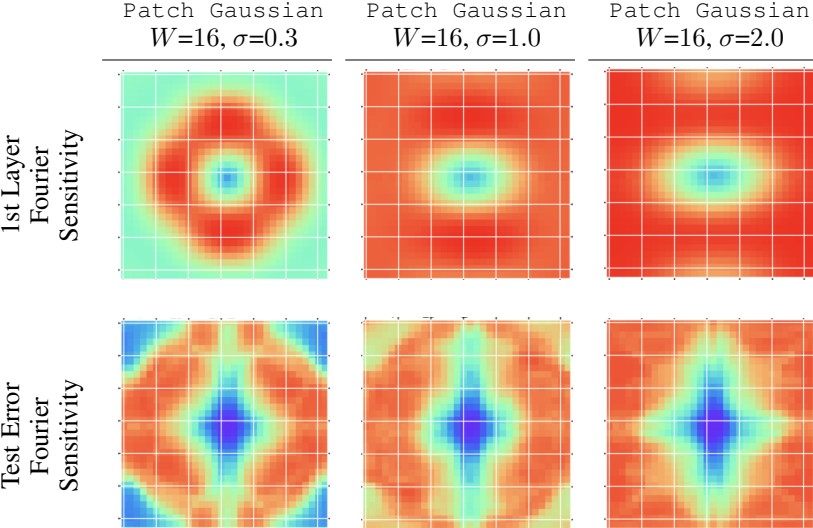

Figure 13: Frequency-based analysis (Yin et al., 2019) for models with different hyper-parameters of `Patch Gaussian`.

