# OpenReview forum: "Improving Robustness Without Sacrificing Accuracy with Patch Gaussian Augmentation"
_ICLR.cc/2020/Conference — Reject_

### Official Review · AnonReviewer3 · 2019-10-23
**Official Blind Review #3**

**Rating:** 3

**Review:**

This paper proposes a hybrid approach for adding noise to training images of an image classification model. Instead of either cutting out a patch or adding gaussian noise, the authors propose to adding a patch of gaussian noise to the images. Although possibly useful practically, this proposal lacks theoretical base on how and why it would be better, besides the claim that hopefully the combination will combine the benefit and subtract the weakness. The experiments are rather limitted to support the claim.

**Experience Assessment:**

I do not know much about this area.

**Review Assessment: Checking Correctness Of Derivations And Theory:**

I assessed the sensibility of the derivations and theory.

**Review Assessment: Checking Correctness Of Experiments:**

I assessed the sensibility of the experiments.

**Review Assessment: Thoroughness In Paper Reading:**

I made a quick assessment of this paper.

---

> ### Author Response · Authors · 2019-11-12
> **Response**
>
> > Although possibly useful practically
>
> We thank the reviewer for pointing out the practical applications of our method. Indeed, because it is so simple, “the approach could become one of the standard mechanisms for data augmentation in the toolset of a practical ML engineer,” as R1 puts it.
>
> > this proposal lacks theoretical base on how and why it would be better
>
> We grant that our work started from an empirical observation. However, we provided an experimental analysis to gain a better understanding of why it works. In particular, Section 5.1 shows that Patch Gaussian seems to allow high-frequency information through at lower layers, but still encourages relatively lower test error sensitivity at high frequencies. Indeed, when we measure accuracy on images filtered with a high-pass filter, we see that Patch Gaussian models can maintain accuracy in a similar way to the baseline and to Cutout, where Gaussian fails to. See Figure 5 for full results.
>
> R1 and R2 agree that our Fourier-theoretic analysis is intuitive. In addition, many practically useful techniques, such as Cutout, do not have completely rigorous mathematical analysis.
>
> > The experiments are rather limited to support the claim
>
> We show extensive experiments highlighting how Patch Gaussian is the only method that retains the benefits of Cutout and Gaussian:
> * We characterize a trade-off between robustness and accuracy among two standard data augmentations - Cutout and Gaussian (Section 2.1). Specifically, Cutout improves accuracy on clean test data. Despite this, we find it does not lead to increased robustness. Conversely, training with higher sigma of Gaussian can lead to increased robustness to Gaussian noise, but it also leads to decreased accuracy on clean data. Therefore, any robustness gains are offset by poor overall performance.
> * We show that our method (Patch Gaussian) allows us to interpolate between the two augmentations above (Section 3.1), and to overcome the observed trade-off, yielding models that are robust to unseen corruptions, while also maintaining clean accuracy (Figure 1, Section 4.1). In doing so, it achieves a new state of the art in the Common Corruptions benchmark on CIFAR-C and ImageNet-C. (Section 4.2), which highlights that simple methods such as ours are competitive with complex training schemes designed for robustness.
> * We demonstrate that Patch Gaussian can be combined with other regularization strategies (Section 4.3) and data augmentation policies (Section 4.4), and can improve COCO object detection performance as well (Section 4.5).
> * We perform a frequency-based analysis of models trained with Patch Gaussian and find that they can better leverage high-frequency information in lower layers, while not being too sensitive to them at later ones (Section 5.1)
>
> We are open to suggestions of further experiment proposals that could convince the reviewer of this.

---

> > ### Comment · AnonReviewer3 · 2019-11-13
> > **Regarding experiments**
> >
> > I appreciate the experiments that the authors presented in the paper and summarize in the response. However, these experiments only shows the empirical performance of Patch Gaussian over baselines on some sample datasets. I wish we can design more systematic experiments to know how Patch Gaussian works, such as if we can see the Fourier analysis (Fig 4)  directly affected by the patch params.
> > For example:
> > 1. When increasing patch size from 1 to image size, how does the sensitivity to high frequency changes. Is it monotonous or have a particular shape?
> > 2. When increasing noise intensity, moving from 0 to total cut out. How does the behavior progress?
> > If we have these, then at least empirically, we can claim that Gaussian or Cutout are specific case of Patch Gaussian and have a better understanding of the problem and solutions. Hence it would be more assured and easy for the practitioners to find a good operating point in a new setting.
> > Table 5 and graphs in the supplemental list the choices, but  the affect of them only shows on performance (which can be biased to dataset), not behaviour which is more insightful.
> >
> > Overall, I buy that this may be a good practice that is useful practically. However, I am not convinced that the authors have fulfil the due diligence on proving the correctness and general behaviour of such technique.

---

> > > ### Author Response · Authors · 2019-11-14
> > > **Re: Regarding experiments**
> > >
> > > > these experiments only shows the empirical behavior of Patch Gaussian over baselines on some sample datasets
> > >
> > > We stress that ImageNet and CIFAR are the most studied vision datasets, and for robustness they are the only datasets with standardized benchmarks (ImageNet-C and CIFAR-10-C).
> > >
> > > > I wish we can design an experiments such that we can see the sensitivity with high frequency directly affected by the patch.
> > > > When increasing noise, moving from 0 to total cut out. How the behavior progress?
> > >
> > > While we do not have time to complete this full set of experiments before the rebuttal deadline, we have analyzed the impact of noise and patch size on fourier sensitivity and have added these results to Figure 13 in the Appendix. They demonstrate that the intuition conveyed in the paper around the analysis of the fourier sensitivity plots is accurate and depends on these hyperparameters.
> > >
> > > Specifically, for patch size 16, stdev=1.0 (Fig 13, center), the Patch Gaussian model demonstrates sensitivity very similar to that reported in Figure 4. The main difference being that the smaller patch size (16 here vs 25 in Figure 4) makes the model slightly more sensitive to high frequencies. This makes sense since smaller patch size moves the model further away from a Gaussian-trained one.
> > >
> > > When we make the scale smaller (patch size 16,  stdev 0.3, left), less information is corrupted in the patch, which moves the model farther from the one trained with Cutout (and therefore closer to a Gaussian-trained one). This can be seen in the increased invariance to high frequencies at the first layer, which is reflected in invariance at test error as well.
> > >
> > > If we, instead, make the scale larger (patch size 16, stdev 2.0, right), we move the model closer to the one trained with Cutout. Notice the higher intensity red in the first layer plot, indicating higher sensitivity to high-frequency features. We also see this sensitivity reflected in the test error, which matches the behavior for Cutout-trained models.
> > >
> > > This confirms that the frequency-based analysis of the models is accurate and reflects changes in hyper-parameters of Patch Gaussian. We also note that this methodology for studying model robustness is not novel to our paper, and has previously been validated and published in NeurIPS 2019.

---

### Official Review · AnonReviewer2 · 2019-10-23
**Official Blind Review #2**

**Rating:** 3

**Review:**

This paper proposes a data augmentation method that interpolates between two existing methods (Cutout and Gaussian), for training robust models towards Gaussian and naturally occurring corruptions. The method is shown to improve robustness without sacrificing accuracy on clean data.
Pros:
The proposed method, despite being simple, seems to empirically work well in terms of the mCE criterion evaluated in the experiments. This does support the authors’ claim that current methods haven’t reached the robustness/accuracy tradeoff boundary yet.
Cons:
I’m a bit concerned about the significance of the work though. The method is a straight-forward combination of existing methods, so methodologically the novelty is kind of limited. Hence, I’m expecting more insights from the analysis of the results, to gain more understanding of why it works so well. However, the presentation of the experiments just seems to aim for the best numbers one can get (I’m not certain how significant the numbers are to this field though). A few examples/pictures of success cases (when the method works) and failure cases (when the method doesn’t work), may help readers (I’m not an expert) to better understand the approach and get more intuitions? The frequency analysis seems quite intuitive. It’s obvious that Gaussian filter blocks high-frequency components, and Cutout keeps some original parts of the image which allow high-freq details to be captured. But, considering CIFAR image size is only 32x32, a patch of size 25 is quite large, how much is the method different from plain whole image Gaussian then?


**Experience Assessment:**

I do not know much about this area.

**Review Assessment: Checking Correctness Of Derivations And Theory:**

N/A

**Review Assessment: Checking Correctness Of Experiments:**

I assessed the sensibility of the experiments.

**Review Assessment: Thoroughness In Paper Reading:**

I read the paper at least twice and used my best judgement in assessing the paper.

---

> ### Author Response · Authors · 2019-11-12
> **Response**
>
> We thank the reviewer for the thoughtful comments. We provide some answers to the concerns raised below:
>
> > I’m a bit concerned about the significance of the work though. The method is a straight-forward combination of existing methods, so methodologically the novelty is kind of limited.
>
> We agree that the method presented is very simple. However, we’d like to emphasize that this was done by design. In showing that such a simple method can be competitive with state-of-the-art methods in the robustness literature, we show that complex training schemes may not be necessary for training models robust to unseen distributions. This is, we believe, where the significance of the work stems. Indeed, R1 mentioned that our method “could become one of the standard mechanisms for data augmentation in the toolset of a practical ML engineer,” especially since it’s so easy to try.
>
> > I’m expecting more insights from the analysis of the results, to gain more understanding of why it works so well.
>
> In Section 5.1, we provide an extensive frequency-based analysis and discussion of why Patch Gaussian works well: Patch Gaussian seems to allow high-frequency information through at lower layers, but still encourages relatively lower test error sensitivity at high frequencies. Indeed, when we measure accuracy on images filtered with a high-pass filter, we see that Patch Gaussian models can maintain accuracy in a similar way to the baseline and to Cutout, where Gaussian fails to. See Figure 5 for full results.
>
> We will re-word this section to clarify these insights to future readers.
>
> > A few examples/pictures of success cases (when the method works) and failure cases (when the method doesn’t work), may help readers (I’m not an expert) to better understand the approach and get more intuitions?
>
> We thank the reviewer for the suggestion. We have not examined this but we hope to include it in camera-ready. In particular, we expect that images with higher Brightness will be among the most common errors, since Patch Gaussian slightly increases error (mCE 0.592) in these corruptions with respect to the Baseline (mCE 0.582). (see Table 7 in Appendix).
>
> > It’s obvious that Gaussian filter blocks high-frequency components, and Cutout keeps some original parts of the image which allow high-freq details to be captured
>
> We agree with the reviewer that these insights make intuitive sense. Our work provides a quantitative evaluation of this phenomenon to confirm this intuition. Further, through rigorous frequency-based sensitivity analysis we show that Patch Gaussian is able to retain both the high frequency sensitivity of Cutout and robustness gains of Gaussian augmentation.
>
> > a patch of size 25 is quite large, how much is the method different from plain whole image Gaussian then?
>
> We remind the reviewer that, while the center of the patch needs to be inside the image, the edges can be outside. This means that, with a patch of size 25, 39.55% of the space is covered in expectation for an image of size 32. Depending on the location of the patch, 16.50% the space is covered (minimum) and other 61.04% is covered (maximum).
>
> In addition, our experimental results clearly show that patch Gaussian performs significantly differently from adding Gaussian noise to the whole image. For example, as shown in Table 1 in our paper, for a Resnet-50 model on ImageNet(-C), Patch Gaussian gets a clean test accuracy of 76% and mCE of 0.714, whereas Gaussian data augmentation gets a clean test accuracy of 75.6% and mCE of 0.739.

---

> ### Public Comment · ~Dan_Hendrycks1 · 2019-12-31
> **RE: Why not submerge the image in Gaussian noise?**
>
> > But, considering CIFAR image size is only 32x32, a patch of size 25 is quite large, how much is the method different from plain whole image Gaussian then?
>
> If the patch were 32x32, or that the whole image was noisy, then the convnet would never see an image with usual local image statistics. Consequently, clean data becomes out-of-distribution or unforeseen during test time, which is not desirable. With a patch size strictly smaller than the whole image, the network can learn how to respond to noisy images and also usual images.

---

### Official Review · AnonReviewer1 · 2019-10-27
**Official Blind Review #1**

**Rating:** 8

**Review:**

The paper proposes a novel data augmentations approach that improves the robustness of a model on the CIFAR-10 and ImageNet Common Corruptions benchmarks while maintaining training accuracy on clean data. To achieve this, the paper proposes a rather simple augmentation mechanism that is inspired by CutOut (DeVries & Taylor 2017) and Gaussian (Grandvalet & Kanu, 1997): adding Gaussian noise to random patches in the image. This simple approach is shown to work surprisingly well on the corruption benchmarks. It seems reasonable that while adding Gaussian noise makes the model robust to high frequency noise, since Gaussian noise is not added everywhere, the model is able to exploit high frequency signal when available in the input. The paper is reasonably well written and the experimental validation is convincing.

Overall, the approach could become one of the standard mechanisms for data augmentation in the toolset of a practical ML engineer.


**Experience Assessment:**

I have read many papers in this area.

**Review Assessment: Checking Correctness Of Derivations And Theory:**

N/A

**Review Assessment: Checking Correctness Of Experiments:**

I carefully checked the experiments.

**Review Assessment: Thoroughness In Paper Reading:**

I read the paper thoroughly.

---

> ### Author Response · Authors · 2019-11-12
> **Response**
>
> We thank the reviewer for the positive comments and helpful summary of our contributions. In particular, we appreciate the summary of the insights demonstrated with the frequency-based analysis (Section 5.1). We hope to incorporate a version of this summary in the camera-ready version as we believe it will be valuable to future readers.

---

### Decision · Program_Chairs · 2019-12-19

**Decision:**

Reject

**Comment:**

The paper in its current form was just not well enough received by the reviewers to warrant an acceptance rating. It seems this work may have promise and the authors are encouraged to continue with this line of work.